# Combined Effects of Rice Husk Biochar and Organic Manures on Soil Chemical Properties and Greenhouse Gas Emissions from Two Different Paddy Soils

**War War Mon [1], Yo Toma [2] and Hideto Ueno [1,\*]**

[1] Department of Bioresource Production Science, United Graduate School of Agriculture, Ehime University, 3-5-7 Tarumi, Matsuyama 790-8566, Japan; 8warwarmonyau@gmail.com

[2] Research Group of Bioscience and Chemistry, Research Faculty of Agriculture, Hokkaido University, Sapporo 060-8589, Japan; toma@agr.hokudai.ac.jp

\* Correspondence: uenoh@agr.ehime-u.ac.jp; Tel.: +81-89-946-9808

**Abstract:** The application of biochar is considered an alternative amendment strategy for improving soil fertility. In this study, we performed pot experiments using soils of low and medium fertility to assess the effects of different combinations of biochar and organic manure on the chemical properties of paddy rice soils and determined the best combination to improve the grain yield without increasing $N_2O$ and $CH_4$ emissions. The applied treatments were without biochar (control), the application of rice husk biochar alone (5 and 10 t ha$^{-1}$), and biochar combined with chicken or cow manure. The results indicated that for both soils, the application of 5 t ha$^{-1}$ biochar combined with 5 t ha$^{-1}$ chicken manure increased grain yield by improving soil total nitrogen and soil $NH_4^+$-N without increasing cumulative $N_2O$ and $CH_4$ emissions. Multiple regression analysis showed that when combined with biochar, chicken manure significantly contributed to a higher grain yield and was negatively associated with cumulative $CH_4$, $N_2O$ emissions, and total GWP. Furthermore, regardless of soil type, combined applications of biochar and cow manure promoted significant increases in soil available P. Our findings indicate that the C/N ratio of organic manure influences $CH_4$ fluxes, and soil type was identified as a factor driving greenhouse gas emissions.

**Keywords:** rice husk biochar; organic manure; soil chemical properties; greenhouse gas emissions; rice (*Oryza sativa*)

## 1. Introduction

Approximately half the global population consumes rice (*Oryza sativa* L.) as a staple food, supplying more than 20% of its total calories [1]. Worldwide, more than 100 countries grow paddy rice, among which Asian countries, including China, India, Indonesia, Bangladesh, Vietnam, Thailand, Myanmar, Japan, the Philippines, Korea, and Pakistan, account for 90% of global rice production [2]. The intensive cultivation of rice in certain Asian regions heavily depends on the input of chemical fertilizers, particularly nitrogen-based fertilizers [3,4]. However, plants have been established to absorb less than half of the applied nitrogen, with most of the remainder being lost to the environment [5]. Not only do these losses contribute to increases in cultivation costs, but they also exacerbate greenhouse gas emissions [6]. In this latter regard, rice cultivation has been estimated to account for 20% of agriculturally derived methane ($CH_4$) and 10% of nitrous oxide ($N_2O$) emissions [7,8]. Therefore, rice production should be guaranteed to improve crop nutrient-use efficiency without increasing fertilizer input for sustainable production and environmental protection.

Biochar, a product obtained from the thermal degradation (pyrolysis) of heterogeneous feedstocks, has been widely documented to have several notable beneficial properties, including energy production, sustainable waste recycling, carbon sequestration, soil quality improvement, and plant growth enhancement [9,10]. One potential biochar feedstock is rice

husks, a byproduct of rice production that is generally wasted (e.g., burned) [11]. However, converting rice husks to biochar could achieve sustainable rice production and effective residue management [12]. Rice husk biochar typically contains high levels of carbon, along with phosphorus, calcium, and magnesium [13], whereas its high surface area is conducive to the colonization of large populations of beneficial microbes and can enhance nutrient retention [14].

As agricultural inputs, organic manures can contribute to enhancing the physical and chemical properties of soil, primarily by reducing soil bulk density and improving soil structure [15]. Although livestock manures are commonly used as organic fertilizers, they tend to be characterized by variable nutrient compositions. For example, whereas chicken manure is a nutrient-rich organic waste containing large amounts of nitrogen, phosphorus, and potassium [16], cow manure has a good balance of nutrients and the potential to serve as a source of phosphorus [17]. In addition, livestock production may differ depending on regions, meat demand options, and food culture [13,15], which can accordingly determine the abundance, commercial availability, and utilization of livestock manures. For instance, in Japan, approximately 70% of dairy cow waste is composted and used in the cultivation of crops and forage plants [18], whereas farmers in Myanmar have long used cow dung to restore soil fertility [19]. Furthermore, in some Asian countries, such as Malaysia, chicken is considered a second staple food, and consequently, chicken manure is particularly abundant [16]. Accordingly, the application of different types of organic manure in agriculture and their respective effects on crop growth should be investigated.

In the context of the aforementioned considerations, it has been established that combining biochar and organic fertilizers can improve soil fertility [20]. However, given that the application of organic materials may contribute to increases in greenhouse gas emissions, effective measures are necessary to simultaneously enhance rice production and reduce greenhouse gas emissions. Currently, however, little information is available regarding the effects of the combined application of biochar and organic manure on soil chemical properties and greenhouse gas emissions. Accordingly, in this study, we conducted pot experiments to (1) determine the changes in soil chemical properties and (2) investigate the optimal combinations of rice husk biochar and different organic manures with respect to their effects on $N_2O$ and $CH_4$ emissions.

## 2. Materials and Methods

### 2.1. Experimental Site and Design

This study was conducted from 13 June to 17 September 2022, under greenhouse conditions at Ehime University, Matsuyama City, Ehime Prefecture, Japan.

Pot experiments were conducted using soils of two fertility types, namely, low- (LF) and medium (MF)-fertility soils. It has been widely documented that soil fertility can influence rice yield. We hypothesized that soil fertility can also affect nutrient uptake, rice growth, and greenhouse gas emissions. To address this consideration, we used two different soil fertility measures. The chemical properties of the two soil types are shown in Table 1. The MF soil was obtained by mixing rice nursery soil (Iseki&Co. Ltd., Ehime, Japan) and sand in a 1:1 ratio, whereas the LF soil was collected from a mountainous area in Toon City, Ehime Prefecture, Japan.

Rice plants (*Oryza sativa* L.) cultivar Koshihikari were transplanted into Wagner pots ($0.02 \text{ m}^2$), with three seedlings being planted in each pot. As soil amendments, we used commercially available rice husk biochar and composted chicken and cow manures. Rice husk biochar has the following properties: pH 6.45, electrical conductivity (EC) 856.3 $\mu\text{S cm}^{-1}$, cation exchange capacity (CEC) 25.4 $\text{cmol}_{(c)} \text{ kg}^{-1}$, exchangeable K content 14,959 $\text{mg kg}^{-1}$, exchangeable Mg content 421.2 $\text{mg kg}^{-1}$, exchangeable Ca content 2415 $\text{mg kg}^{-1}$, ash 44.9%, volatile matter 18.7%, and persistent carbon 28.5%. The chicken manure had a total N content of 4.05%, total C content of 25.04%, C/N ratio of 6.19, and available P content of 1334.14 $\text{mg kg}^{-1}$, whereas the cow manure had a total N content of 1.88%, total C content of 34.09%, C/N ratio of 18.14, and available P content of 2548 $\text{mg kg}^{-1}$. The

application rates of biochar, chicken manure, and cow manure were determined in line with Japanese recommendations and based on considerations of price and economic feasibility. The biochar was applied at 5 t ha$^{-1}$ (10 g pot$^{-1}$) and 10 t ha$^{-1}$ (20 g pot$^{-1}$), whereas both chicken and cow manure were applied at 5 t ha$^{-1}$ (10 g pot$^{-1}$ on a fresh weight basis). For each of the two soils, we performed the following seven treatments: (i) control (without biochar) (C), (ii) 5 t ha$^{-1}$ rice husk biochar (B5), (iii) 10 t ha$^{-1}$ rice husk biochar (B10), (iv) 5 t ha$^{-1}$ rice husk biochar + 5 t ha$^{-1}$ chicken manure (B5:CHM), (v) 5 t ha$^{-1}$ rice husk biochar + 5 t ha$^{-1}$ cow manure (B5:COM), (vi) 10 t ha$^{-1}$ rice husk biochar + 5 t ha$^{-1}$ chicken manure (B10:CHM), and (vii) 10 t ha$^{-1}$ rice husk biochar + 5 t ha$^{-1}$ cow manure (B10:COM).

**Table 1.** Chemical properties of low- and medium-fertility soils prior to cultivation.

| Measurements | Unit | MF | LF |
|---|---|---|---|
| pH | | 6.17 | 7.86 |
| Electrical conductivity | µS cm$^{-1}$ | 274.0 | 21.0 |
| Exchangeable K content | cmol$_{(c)}$ kg$^{-1}$ | 0.15 | 0.06 |
| Exchangeable Mg content | cmol$_{(c)}$ kg$^{-1}$ | 1.08 | 0.88 |
| Exchangeable Ca content | cmol$_{(c)}$ kg$^{-1}$ | 10.3 | 9.84 |
| Total N content | % | 0.18 | 0.02 |
| Total C content | % | 0.53 | 0.03 |
| C/N | | 2.90 | 1.68 |
| NH$_4^+$-N content | mg kg$^{-1}$ | 26.7 | 11.7 |
| NO$_3^-$-N content | mg kg$^{-1}$ | 0.46 | 0.23 |
| Available P content | mg kg$^{-1}$ | 54.6 | 43.0 |

MF: Medium fertility soil, LF: low fertility soil.

### 2.2. Soil Media Preparation and Fertilization

Each of the experimental pots was filled with 4 kg of air-dried soil. One week prior to rice seedling transplantation, rice husk biochar and organic manures were thoroughly mixed into all layers of the potting soil. Given the relatively low fertility of the experimental soils, N-P-K (15%-15%-15%) fertilizer was split-applied, with 0.53 g pot$^{-1}$ as a basal fertilizer, followed by subsequent applications of 0.40 g pot$^{-1}$ at 14 and 30 days after transplanting (DAT). Thus, all pots received a total of 1.33 g pot$^{-1}$ of fertilizer. In addition, for all treatments, supplemental urea fertilization was performed as a top dressing of 0.13 gN pot$^{-1}$ (30 kgN ha$^{-1}$) at 40, 47, and 54 DAT. The nutrients received from the organic manures and supplemental fertilizers for each treatment are listed in Table 2. All pots also received daily irrigation. The paddy rice was cultivated for a period of 96 days, from 13 June to 17 September.

**Table 2.** The nutrients received in each treatment from chicken manure, cow manure, and supplemental fertilizers.

| Treatments | Carbon (g C pot$^{-1}$) | Nitrogen (g N pot$^{-1}$) | Phosphorus (g P$_2$O$_5$ pot$^{-1}$) | Potassium (g K$_2$O pot$^{-1}$) |
|---|---|---|---|---|
| C | - | 0.38 | 0.20 | 0.20 |
| B5 | - | 0.38 | 0.20 | 0.20 |
| B10 | - | 0.38 | 0.20 | 0.20 |
| B5:CHM | 25.0 | 0.78 | 0.21 | 0.56 |
| B5:COM | 34.1 | 0.57 | 0.22 | 0.30 |
| B10:CHM | 25.0 | 0.78 | 0.21 | 0.56 |
| B10:COM | 34.1 | 0.57 | 0.22 | 0.30 |

C: Control, B5: 5 t ha$^{-1}$ rice husk biochar, B10: 10 t ha$^{-1}$ rice husk biochar, B5:CHM: 5 t ha$^{-1}$ rice husk biochar + 5 t ha$^{-1}$ chicken manure, B5:COM: 5 t ha$^{-1}$ rice husk biochar + 5 t ha$^{-1}$ cow manure, B10:CHM: 10 t ha$^{-1}$ rice husk biochar + 5 t ha$^{-1}$ chicken manure, B10:COM: 10 t ha$^{-1}$ rice husk biochar + 5 t ha$^{-1}$ cow manure.

The chemical properties of the soils were analyzed before and after cultivation. Prior to analyses, collected soil samples were air-dried, ground, and sieved ($\leq 2$ mm). Soil pH was determined from soil–water suspensions (1:2.5, $v/v$) using a B-212 pH meter (HORIBA, Kyoto, Japan), and EC was determined using a B173 Horiba Twin Cond Conductivity Meter. Soil $NH_4^+$-N and $NO_3^-$-N were extracted with 2 M KCl, and their concentrations were determined calorimetrically using the indophenol blue and vanadium chloride nitrate reduction methods, respectively. Total C and N contents were analyzed by the dry combustion method using a Vario Max CN elemental analyzer (Elementar Analysensysteme GmbH, Germany), and available P content was measured using the Bray II method.

### 2.3. Measurement of Cumulative $CH_4$ and $N_2O$ Emissions

$CH_4$ and $N_2O$ gas samples were collected at 1 and 6 days after fertilizer application and at 2, 7, 14, 21, 35, 49, 63, 77, and 91 DAT, using the closed-chamber method. The acrylic chambers used were equipped with a fan, thermometer, and tube for sample collection. Gas samples were collected at 0, 10, and 20 min after the chamber was installed. At the early stage of rice cultivation, from 2 DAT to 21 DAT, we used a short acrylic chamber with a diameter of 16 cm and height of 16 cm, whereas from 35 DAT to 91 DAT, we used a tall acrylic chamber with a diameter of 16 cm and height of 85 cm. Gas samples were taken by inserting a 20 mL syringe needle through the chamber sample collection tube, which was then injected into vacuum-sealed vials fitted with butyl rubber stoppers. The concentrations of $CH_4$ and $N_2O$ in the gas samples were simultaneously analyzed using a gas chromatograph equipped with flame ionization and electron capture detectors (GC-14A, Shimadzu, Kyoto, Japan). $CH_4$ and $N_2O$ fluxes were calculated using linear regression, and the cumulative fluxes were determined using the trapezoidal method proposed by Toma et al. [21] as follows:

$$F = \rho \times V/A \times dC/dt \times [273/(273 + T)] \times \alpha, \tag{1}$$

where F is the flux (mg m$^{-2}$ h$^{-1}$), $\rho$ is the density of $CH_4$ and $N_2O$ at standard temperature and pressure (0.717 mg m$^{-3}$ for $CH_4$ and 1.97 mg m$^{-3}$ for $N_2O$), $V$ is the volume of the chamber (m$^3$), $A$ is the cross-sectional area of the chamber (m$^2$), $dC/dt$ is the ratio of change in the gas concentrations within the chamber per unit time, $T$ is the average air temperature within the chamber (°C), and $\alpha$ is the conversion factor of $CH_4$ to C (12/16) or $N_2O$ to N (28/44). The trapezoidal rule was used to calculate the cumulative emissions of $CH_4$ and $N_2O$.

### 2.4. Global Warming Potential

Global warming potential (GWP) is an indicator of the impact of greenhouse gases on global warming and is typically calculated by using $CO_2$ as a reference gas and converting changes in $CH_4$ or $N_2O$ emissions to $CO_2$-equivalents. On the basis of the GWP of $CO_2$ over a 100-year time horizon, the GWP for $CH_4$ is 34, and that for $N_2O$ is 298 [22]. GWP was calculated using the following equations:

$$GWP_{CH4} = \text{cumulative } CH_4 \text{ emissions (MgC ha}^{-1} \text{ period}^{-1}) \times 16/12 \times 34 \tag{2}$$

$$GWP_{N2O} = \text{cumulative } N_2O \text{ emissions (MgN ha}^{-1} \text{ period}^{-1}) \times 44/28 \times 298 \tag{3}$$

The total global warming potential period$^{-1}$ ($GWP_{total}$) was obtained as the sum of $GWP_{CH4}$ and $GWP_{N2O}$.

### 2.5. Statistical Analysis

Real Statistics Resource Pack software (Release 8.4) was used to analyze the data. Two-way factorial analysis of variance (ANOVA) was used to determine whether plant growth and $CH_4$ and $N_2O$ emissions were affected by biochar application, organic fertilizer application, or their interaction in the two soils. Multiple comparisons among the treatment

means were performed using Tukey's honest significant difference test at a significance level of $p < 0.05$.

## 3. Results

### 3.1. Effects of Biochar and Different Organic Matter Additives on Rice Dry Biomass and Grain Yield

The mean values of the aboveground dry biomass, root dry biomass, and grain yield of the rice are shown in Table 3. In the case of the MF soil, with respect to aboveground biomass, we detected no significant differences among the treatments in which biochar was applied alone or with organic manures, with the maximum value being obtained in the B5:CHM treatment. In terms of root dry biomass, we detected no significant differences among the B5, B10, B5:CHM, B5:COM, and B10:CHM treatments, whereas in contrast, the B10:COM and control treatments were characterized by significantly lower values. Collectively, these findings revealed that in the MF soil, the highest aboveground and root dry biomasses were obtained in response to the B5:CHM treatment. In terms of grain yield, values arranged from the highest to the lowest were obtained for treatments as follows: B5:CHM > B10:CHM > B5 = B10 > B10:COM > B5:COM > C. The highest grain yield of 32.5 g pot$^{-1}$ was obtained in response to the B5:CHM treatment and was found to be significantly higher than the yields obtained with other treatments. Comparatively, the control (without biochar) treatment gave the lowest grain yield.

**Table 3.** The aboveground dry biomass, root dry biomass, and grain yield of rice under different treatments in low- and medium-fertility soils (mean ± standard error).

| Treatments | Aboveground Dry Biomass (g pot$^{-1}$) | | Root Dry Biomass (g pot$^{-1}$) | | Grain Yield (g pot$^{-1}$) | |
|---|---|---|---|---|---|---|
| | MF | LF | MF | LF | MF | LF |
| C | 38.9 ± 3.5 b | 10.9 ± 0.4 c | 4.74 ± 0.2 b | 2.72 ± 0.3 b | 18.9 ± 1.0 c | 9.9 ± 0.9 d |
| B5 | 53.6 ± 1.8 a | 22.3 ± 1.4 a | 5.14 ± 0.2 a | 3.10 ± 0.3 a | 25.2 ± 0.8 b | 22.1 ± 1.0 b |
| B10 | 52.4 ± 4.5 a | 21.5 ± 2.1 b | 5.80 ± 0.4 a | 3.68 ± 0.2 a | 25.2 ± 0.7 b | 19.5 ± 1.4 c |
| B5:CHM | 58.1 ± 4.1 a | 28.5 ± 0.6 a | 7.12 ± 0.9 a | 4.28 ± 0.3 a | 32.5 ± 1.9 a | 27.5 ± 0.7 a |
| B5:COM | 47.9 ± 1.8 a | 22.2 ± 1.0 a | 5.20 ± 0.3 a | 3.62 ± 0.2 a | 22.9 ± 0.8 b | 19.4 ± 0.5 bc |
| B10:CHM | 56.4 ± 1.7 a | 30.2 ± 3.9 a | 7.16 ± 0.7 a | 3.88 ± 0.5 a | 25.3 ± 2.7 b | 24.8 ± 0.3 a |
| B10:COM | 45.9 ± 2.0 a | 21.0 ± 2.0 ab | 4.68 ± 0.2 c | 3.94 ± 0.5 a | 25.1 ± 0.6 b | 18.8 ± 2.1 bc |
| Between two soils | <0.001 | | <0.001 | | <0.001 | |
| Within treatments | <0.001 | | <0.001 | | <0.001 | |
| Soils × Treatments | 0.79 | | 0.09 | | 0.06 | |

Different letters following values within the same column denote significant differences between groups at the 5% level, as determined using the LSD test. MF: medium-fertility soil; LF: low-fertility soil. C: Control, B5: 5 t ha$^{-1}$ rice husk biochar, B10: 10 t ha$^{-1}$ rice husk biochar, B5:CHM: 5 t ha$^{-1}$ rice husk biochar + 5 t ha$^{-1}$ chicken manure, B5:COM: 5 t ha$^{-1}$ rice husk biochar + 5 t ha$^{-1}$ cow manure, B10:CHM: 10 t ha$^{-1}$ rice husk biochar + 5 t ha$^{-1}$ chicken manure, B10:COM: 10 t ha$^{-1}$ rice husk biochar + 5 t ha$^{-1}$ cow manure.

With respect to the low-fertility soil, the highest aboveground dry biomass was obtained in response to the B10:CHM treatment, although statistically similar values were recorded for the B10:CHM, B5:CHM, B5, and B10:COM treatments. Contrastingly, the highest root dry biomass was obtained in soils amended with B5:CHM, although apart from the control treatment, we detected no significant differences among treatments with respect to this parameter. Treatment-wise, the values obtained for grain yield could be ordered as follows: B5:CHM > B10:CHM > B5 > B10 > B5:COM > B10:COM > C. Although a maximum grain yield of 27.5 g pot$^{-1}$ was recorded for plants receiving the B5:CHM treatment, this value did not differ significantly from the 24.8 g pot$^{-1}$ obtained for those treated with B10:CHM. Similar to the MF soil, the lowest values for the production of aboveground dry biomass, root dry biomass, and grain yield were obtained for rice plants receiving the control treatment. Overall, however, higher grain yields were obtained from plants transplanted into MF soil. For neither of the two soil types did we detect any significant

interactions between the soils and the different treatments with respect to dry biomass production or grain yield.

Very small differences among the treatments were observed with the combined application of rice husk biochar and organic manures, without a significant difference in soil pH, after harvest in both soils (see Table 4). Comparatively, soil pH increased significantly in soils amended with the combined application of biochar and organic manures relative to the control and biochar applications alone in both soils. In MF soil, relative to the original soil, we recorded post-harvest increases of 3.9%, 2.9%, 9.7%, 6.2%, 13.5%, and 14.7% in the pH of soils receiving the B5, B10, B5:CHM, B5:COM, B10:CHM, and B10:COM treatments, respectively. Conversely, we recorded a 0.32% reduction in the pH of the soil under the control treatment in MF soil. In the case of low-fertility soil, the original soil had a relatively high pH (7.86), and consequently, for all treatments, we detected reductions in soil pH after rice cultivation by 20.1%, 14.4%, 12.31%, 7.76%, 10.8%, 11.9%, and 7.5% in the control, B5, B10, B5:CHM, B5:COM, B10:CHM, and B10:COM treatments, respectively. Notably, for both soil types, soil amended with B10:COM was found to have the highest pH.

**Table 4.** Soil pH, EC, and total N after cultivation (means ± standard error).

| Treatments | Soil pH | | Soil EC ($\mu$S cm$^{-1}$) | | Total N Content (%) | |
|---|---|---|---|---|---|---|
| | MF | LF | MF | LF | MF | LF |
| C | 6.15 ± 0.05 d | 6.28 ± 0.03 d | 530.0 ± 20.0 d | 103.5 ± 7.5 d | 0.12 ± 0.005 b | 0.029 ± 0.002 b |
| B5 | 6.41 ± 0.04 b | 6.73 ± 0.07 c | 682.5 ± 7.5 b | 122.5 ± 7.5 c | 0.12 ± 0.001 b | 0.022 ± 0.001 c |
| B10 | 6.35 ± 0.05 c | 6.89 ± 0.09 b | 653.5 ± 13.5 c | 162.5 ± 7.5 b | 0.17 ± 0.001 b | 0.034 ± 0.002 b |
| B5:CHM | 6.77 ± 0.14 a | 7.25 ± 0.15 a | 715.0 ± 5.0 a | 285.0 ± 5.0 a | 0.22 ± 0.015 a | 0.036 ± 0.003 b |
| B5:COM | 6.55 ± 0.06 a | 7.01 ± 0.10 a | 705.0 ± 7.0 a | 170.5 ± 19.5 a | 0.16 ± 0.001 b | 0.030 ± 0.002 b |
| B10:CHM | 7.00 ± 0.10 a | 6.92 ± 0.09 a | 765.0 ± 15.0 a | 275.0 ± 25.0 a | 0.15 ± 0.015 b | 0.045 ± 0.003 a |
| B10:COM | 7.08 ± 0.22 a | 7.27 ± 0.04 a | 730.0 ± 20.0 a | 228.5 ± 11.5 a | 0.14 ± 0.015 b | 0.033 ± 0.002 b |
| Between two soils | <0.001 | | <0.001 | | <0.001 | |
| Within treatments | <0.001 | | <0.001 | | <0.001 | |
| Soils × Treatments | 2.22 | | 0.002 | | <0.001 | |

Different letters following values within the same column denote significant differences between groups at the 5% level, as determined using the LSD test. MF: medium-fertility soil; LF: low-fertility soil. C: Control, B5: 5 t ha$^{-1}$ rice husk biochar, B10: 10 t ha$^{-1}$ rice husk biochar, B5:CHM: 5 t ha$^{-1}$ rice husk biochar + 5 t ha$^{-1}$ chicken manure, B5:COM: 5 t ha$^{-1}$ rice husk biochar + 5 t ha$^{-1}$ cow manure, B10:CHM: 10 t ha$^{-1}$ rice husk biochar + 5 t ha$^{-1}$ chicken manure, B10:COM: 10 t ha$^{-1}$ rice husk biochar + 5 t ha$^{-1}$ cow manure.

With respect to soil EC after harvest in the MF soil, the recorded values obtained in the different treatments descended in the order B10:CHM > B10:COM > B5:CHM > B5:COM > B5 > B10 > C, whereas for the LF soil, the corresponding trend was B5:CHM > B10:CHM > B10:COM > B5:COM > B10 > B5 > C. It is evident that the difference was slight among the combinations of biochar and organic manure, but they were significantly higher than those of the control and biochar application alone in both soils. We also detected an interaction effect between soil and treatment with respect to soil EC. Furthermore, we established that for the MF soil, the B5:CHM combined treatment was associated with the highest total nitrogen, whereas the B10:CHM treatment produced the highest total nitrogen content in the LF soil. Moreover, an interaction effect between treatment and soil type was observed.

### 3.2. Changes in Soil Chemical Properties following Rice Cultivation

Our analyses of soil chemical properties revealed that in both soil types, there was a significant increase in the available NH$_4^+$-N content in the soil in response to application of the B5:CHM combination treatment (Figure 1), with treatment-wise values descending in the order B5:CHM > B10 > B10:CHM > B5 > B10:COM > B5:COM > C for the MF soil and B5:CHM > B10:CHM > B10 > B5 > B10:COM > B5:COM > C for the LF soil. Accordingly, for both soil types, the combined application of 5 t ha$^{-1}$ rice husk biochar and 5 t ha$^{-1}$ chicken manure was found to produce the highest soil available NH$_4^+$-N content.

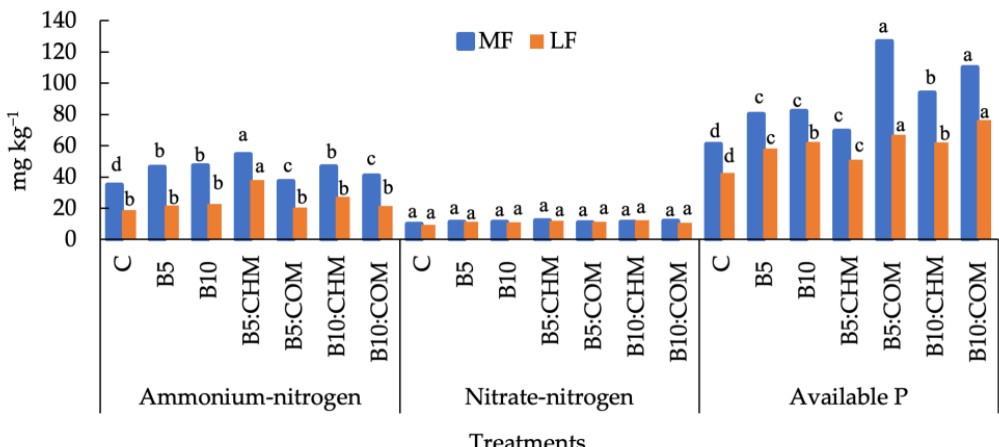

**Figure 1.** Soil ammonium–nitrogen, nitrate–nitrogen, and available P contents under different treatments in MF and LF soil after harvest. Different letters represent statistically significant differences at the 5% level, as determined using the LSD test. MF: medium-fertility soil; LF: low-fertility soil. C: Control, B5: 5 t ha$^{-1}$ rice husk biochar, B10: 10 t ha$^{-1}$ rice husk biochar, B5:CHM: 5 t ha$^{-1}$ rice husk biochar + 5 t ha$^{-1}$ chicken manure, B5:COM: 5 t ha$^{-1}$ rice husk biochar + 5 t ha$^{-1}$ cow manure, B10:CHM: 10 t ha$^{-1}$ rice husk biochar + 5 t ha$^{-1}$ chicken manure, B10:COM: 10 t ha$^{-1}$ rice husk biochar + 5 t ha$^{-1}$ cow manure.

As indicated in Figure 1, no significant differences were observed among the assessed treatments with respect to soil $NO_3^-$-N content, with obtained values descending in the order B5:CHM > B10:COM > B10:CHM > B10 > B5 > B5:COM > C for the MF soil, and B10:CHM > B5:CHM > B10 > B5:COM > B10 > B10:COM > C for the LF soil. For the MF soil, the highest $NO_3^-$-N content (12.09 mg kg$^{-1}$) was detected in soil receiving the B5:CHM treatment, whereas for the LF soil, the highest value (12.23 mg kg$^{-1}$) was obtained in response to the B10:CHM treatment. For both soils, the lowest values were obtained in response to the respective control treatments, measured at 10.0 and 9.5 mg kg$^{-1}$ for the MF and LF soils, respectively.

As shown in Figure 1, incorporating biochar and cow manure into the soil significantly increased the available $P_2O_5$ content. For MF soil, the recorded values of soil available P content descended in the order B5:COM > B10:COM > B10:CHM > B10 > B5 > B5:CHM > C, with values of 126.9, 110.2, 93.9, 82.1, 80.1, 69.4, and 61.1 mg kg$^{-1}$, respectively. Comparatively, in LF soil, contents descended in the order B10:COM > B5:COM > B10 > B10:CHM > B5 > B5:CHM > C, with corresponding values of 76.62, 67.1, 62.3, 62.1, 58.4, 51.2, and 42.5 mg kg$^{-1}$.

### 3.3. CH$_4$ and N$_2$O Fluxes from Low- and Medium-Fertility Soils

The CH$_4$ and N$_2$O fluxes from both soils under the different treatments are shown in Figure 2. In the case of the MF soil, we detected only low CH$_4$ fluxes prior to 20 DAT, after which we observed notable fluctuations, with an initial small peak being observed at 35 DAT. Thereafter, we detected a markedly increased fluctuation, with the highest peak being recorded at 76 DAT (1205.9 µgC m$^{-2}$ h$^{-1}$) in soil receiving the B5:COM treatment (Figure 2a). Comparatively, soil amended with B5:CHM showed the lowest flux (−114.6 µgC m$^{-2}$ h$^{-1}$), recorded at 51 DAT. Similar fluctuations in CH$_4$ were observed in the LF soil, with the highest peak being detected at 76 DAT (1184.9 µgC m$^{-2}$ h$^{-1}$) in soil amended with B10:COM (Figure 2b), and the lowest flux was observed at 35 DAT in soil receiving the B5 treatment (−194.8 µgC m$^{-2}$ h$^{-1}$). For both soils, however, we found only small CH$_4$ fluctuations during the vegetative stage of rice growth.

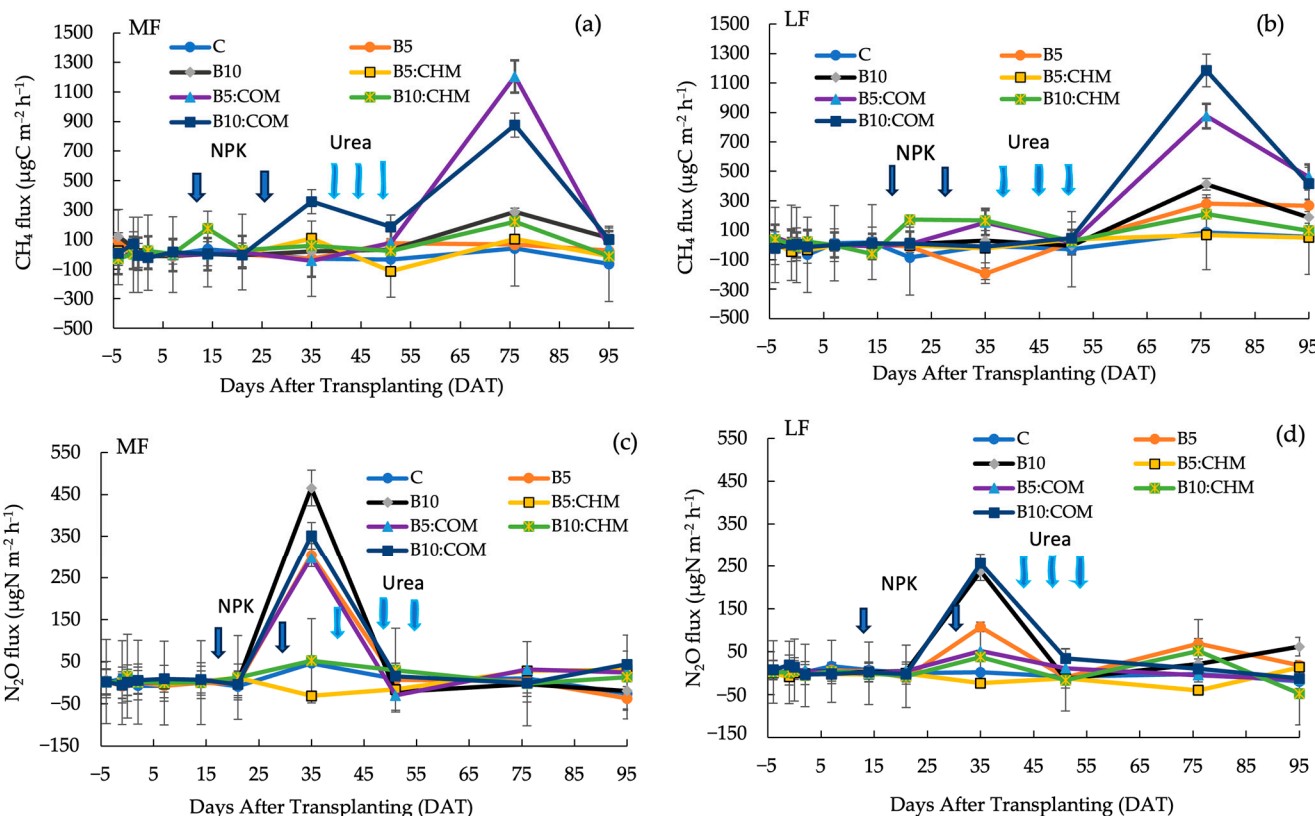

**Figure 2.** The variations in $CH_4$ flux from MF (**a**) and LF (**b**) and $N_2O$ flux from MF (**c**) and LF (**d**) during the experiment. Error bars represent standard errors. MF: medium-fertility soil; LF: low-fertility soil C: Control, B5: 5 t ha$^{-1}$ rice husk biochar, B10: 10 t ha$^{-1}$ rice husk biochar, B5:CHM: 5 t ha$^{-1}$ rice husk biochar + 5 t ha$^{-1}$ chicken manure, B5:COM: 5 t ha$^{-1}$ rice husk biochar + 5 t ha$^{-1}$ cow manure, B10:CHM: 10 t ha$^{-1}$ rice husk biochar + 5 t ha$^{-1}$ chicken manure, B10:COM: 10 t ha$^{-1}$ rice husk biochar + 5 t ha$^{-1}$ cow manure.

The variations in $N_2O$ fluxes measured in MF and LF soils are shown in Figure 2c,d, respectively. For MF soil, the highest $N_2O$ peak was detected at 35 DAT in soil treated with B10 (465.5 µgN m$^{-2}$ h$^{-1}$). Similarly, the $N_2O$ flux peaked at 35 DAT in LF soil, although in response to the B10:COM treatment (255.9 µgN m$^{-2}$ h$^{-1}$). In both soil types, the lowest $N_2O$ fluxes were observed in soil amended with B5:CHM, at 35 DAT in MF soil (−31.1 µgN m$^{-2}$ h$^{-1}$) and 76 DAT in LF soil (−40.1 µgN m$^{-2}$ h$^{-1}$).

### 3.4. Cumulative Emissions of Soil $CH_4$ and $N_2O$

Our findings regarding the cumulative emissions of $CH_4$ and $N_2O$ during the 96-day treatment period are presented in Table 5. The lowest $CH_4$ emission was recorded in C; however, this was not statistically different from the combined application of B5:CHM and B5 in MF. We found that all $CH_4$ emission values were of the same order of magnitude in both soils. The order of $CH_4$ emissions was arranged from lowest to highest: C < B5:CHM < B5 < B10:CHM < B10 < B5:COM < B10:COM. Regarding $N_2O$ emissions, soil amended with B5:CHM was found to be associated with the lowest $N_2O$ emissions during the treatment period, whereas the highest emissions from MF and LF soils were recorded for those soils receiving the B10 and B10:COM treatments, respectively. Notably, except for B5:COM, all treatments contributed to a significantly larger release of $CH_4$ from the LF and MF soils; the opposite trend was observed for the cumulative emissions of soil $N_2O$. However, we detected no significant interactions between treatments and soil type with respect to these emissions.

**Table 5.** Cumulative emissions of $CH_4$ and $N_2O$ (means $\pm$ standard error).

| Treatments | Cumulative $CH_4$ Emission (mg C m$^{-2}$ 96 days$^{-1}$) | | Cumulative $N_2O$ Emission (mg N m$^{-2}$ 96 days$^{-1}$) | |
|---|---|---|---|---|
| | MF | LF | MF | LF |
| C | $-2.6 \pm 6.4$ de | $20.9 \pm 6.3$ e | $28.1 \pm 8.8$ a | $-4.0 \pm 8.0$ a |
| B5 | $67.4 \pm 7.9$ d | $148.2 \pm 14.3$ bc | $101.4 \pm 23.9$ a | $37.0 \pm 31.9$ a |
| B10 | $197.2 \pm 7.0$ b | $271.36 \pm 21.8$ b | $146.9 \pm 29.4$ a | $104.5 \pm 9.5$ a |
| B5:CHM | $36.3 \pm 21.7$ e | $53.76 \pm 6.7$ d | $7.3 \pm 2.1$ b | $-36.5 \pm 27.2$ b |
| B5:COM | $674.7 \pm 24.9$ a | $633.1 \pm 0.3$ a | $113.9 \pm 39.9$ a | $20.0 \pm 15.0$ a |
| B10:CHM | $186.0 \pm 11.3$ c | $245.2 \pm 38.8$ c | $39.1 \pm 33.9$ a | $15.4 \pm 3.3$ a |
| B10:COM | $701.8 \pm 6.4$ a | $742.1 \pm 34.0$ a | $141.6 \pm 5.6$ a | $108.4 \pm 47.0$ a |
| Between two soils | <0.05 | | <0.05 | |
| Within treatments | <0.001 | | <0.001 | |
| Soils $\times$ Treatments | 0.09 | | 0.8 | |

Different letters following values within the same column denote significant differences between groups at the 5% level, as determined using the LSD test. MF: medium-fertility soil; LF: low-fertility soil. C: Control, B5: 5 t ha$^{-1}$ rice husk biochar, B10: 10 t ha$^{-1}$ rice husk biochar, B5:CHM: 5 t ha$^{-1}$ rice husk biochar + 5 t ha$^{-1}$ chicken manure, B5:COM: 5 t ha$^{-1}$ rice husk biochar + 5 t ha$^{-1}$ cow manure, B10:CHM: 10 t ha$^{-1}$ rice husk biochar + 5 t ha$^{-1}$ chicken manure, B10:COM: 10 t ha$^{-1}$ rice husk biochar + 5 t ha$^{-1}$ cow manure.

Figure 3 presents data obtained for the values of GWP$_{total}$ measured during the 96-day treatments, which indicate a similar trend of GWP$_{total}$ in both assessed soil types, arranged from lowest to highest as follows: B5:CHM < C < B10:CHM < B5 < B10 < B5:COM < B10:COM. Accordingly, for both soil types, B10:COM contributed to the highest GWP$_{total}$.

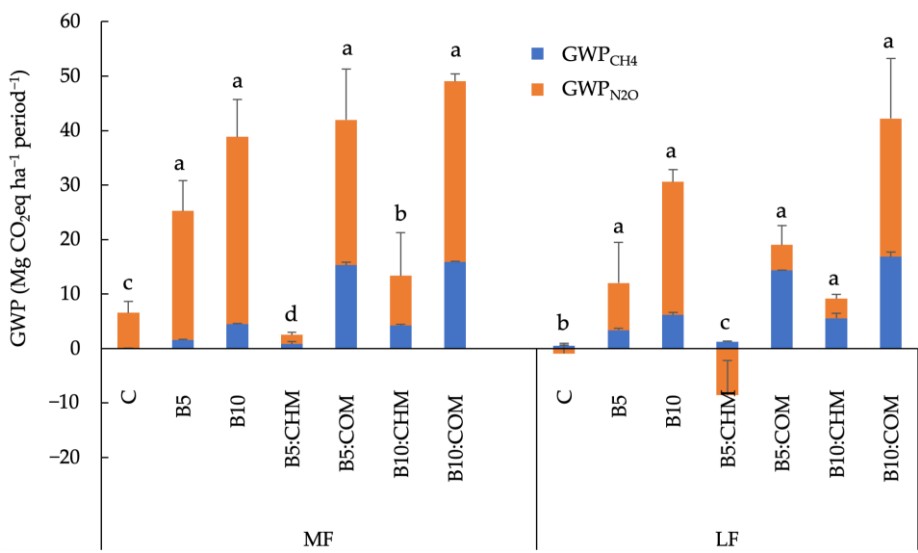

**Figure 3.** The total global warming potential of $CH_4$ (GWP$_{CH4}$) and $N_2O$ (GWP$_{N2O}$) in the pot experiments. All values are expressed as the means $\pm$ standard error. Different letters above the bar denote statistically significant differences at the 5% level, as determined using the LSD test. MF: medium-fertility soil; LF: low-fertility soil. C: Control, B5: 5 t ha$^{-1}$ rice husk biochar, B10: 10 t ha$^{-1}$ rice husk biochar, B5:CHM: 5 t ha$^{-1}$ rice husk biochar + 5 t ha$^{-1}$ chicken manure, B5:COM: 5 t ha$^{-1}$ rice husk biochar + 5 t ha$^{-1}$ cow manure, B10:CHM: 10 t ha$^{-1}$ rice husk biochar + 5 t ha$^{-1}$ chicken manure, B10:COM: 10 t ha$^{-1}$ rice husk biochar + 5 t ha$^{-1}$ cow manure.

*3.5. Effects of Biochar, Organic Manures, and Soil Type on Grain Yield, Greenhouse Gas Emissions, and Global Warming Potential*

To compare the relative importance of the effects of biochar, chicken manure, cow manure, and soil type on grain yield, cumulative $CH_4$ emissions, cumulative $N_2O$ emissions, GWP$_{CH4}$, GWP$_{N2O}$, and GWP$_{total}$, we performed standardized regression analysis (Table 6). On the basis of this analysis, we determined that when combined with biochar,

chicken manure made the most significant contribution to grain yield and showed negative associations with cumulative $CH_4$ and $N_2O$ emissions, $GWP_{CH4}$, $GWP_{N2O}$, and $GWP_{total}$. Soil type was similarly identified as an important factor in promoting grain yield. Notably, the application of cow manure resulted in the highest cumulative $CH_4$ emissions. In addition, the application of biochar was established to have a positive effect on cumulative $N_2O$ emissions. Furthermore, soil type was found to have the greatest positive effect on $GWP_{CH4}$ and biochar application was associated with $GWP_{N2O}$ and $GWP_{total}$.

**Table 6.** Standardized regression coefficients obtained for treatment factors determined using multiple regression analysis.

| Response Valuables | Explanatory Valuables | | | | | |
|---|---|---|---|---|---|---|
| | **Grain Yield** | **Cumulative $CH_4$ Emission** | **Cumulative $N_2O$ Emission** | **$GWP_{CH4}$** | **$GWP_{N2O}$** | **$GWP_{total}$** |
| Biochar | 0.24 * | 0.30 *** | 0.62 *** | 0.42 *** | 0.62 *** | 0.17 |
| Chicken Manure | 0.51 *** | −0.07 * | −0.66 *** | −0.43 * | −0.66 *** | −0.09 |
| Cow Manure | 0.04 | 0.86 *** | 0.13 | 0.25 *** | −0.002 | 0.16 |
| Soil Types | 0.41 *** | −0.07 * | 0.39 *** | 0.51 * | 0.39 *** | 0.04 |

$GWP_{CH4}$: global warming potential of $CH_4$, $GWP_{N2O}$: global warming potential of $N_2O$, $GWP_{total}$: sum of the global warming potential of $GWP_{CH4}$ and $GWP_{N2O}$. *: significant at $p < 0.05$. ***: significant at $p < 0.001$.

## 4. Discussion

### 4.1. Effects of Biochar and Organic Manures on Grain Yield and the Chemical Properties of Paddy Soils

Among the amendment treatments assessed in this study, we found that in a soil of medium fertility, the maximum aboveground biomass of rice was obtained in response to the combined application of 5 t ha$^{-1}$ biochar and 5 t ha$^{-1}$ chicken manure, whereas the application of 10 t ha$^{-1}$ biochar and 5 t ha$^{-1}$ chicken manure yielded the highest aboveground biomass of rice cultivated in low fertility soil. Furthermore, for both soil types, the highest grain yield and root biomass were obtained in response to the combined application of 5 t ha$^{-1}$ biochar and 5 t ha$^{-1}$ chicken manure. Consistently, standardized regression analysis revealed that chicken manure significantly influenced rice grain yield (Table 6). Even though we applied the same amounts of both organic manures and supplied the same amounts of supplemental fertilizers to all treatments, there were differences in the nutrient content of organic manures. In particular, whereas chicken manure had a 4.05% N content, that of cow manure was somewhat lower at 1.88%. Consequently, soils treated with chicken manure (B5:CHM and B10:CHM) would have received 0.78 g N pot$^{-1}$ more nitrogen (Table 2). It is thus conceivable that differences in the nutrient contents of manures had an appreciable influence on the morphological growth and grain yield of rice plants.

In addition to having a higher total N content, the chicken manure used in this study also had a lower C/N ratio compared with that of cow manure. We speculate that when combined with biochar, this property may have contributed to enhancing the accumulation of soil total N and thus nitrogenous nutrients, particularly available $NH_4^+$ (Figure 1). Chicken manure with a low C/N ratio contributed to improving microorganism activity, particularly the enhancement of urease activity, which would be conducive to accelerating the transformation of soil mineral nutrients to the free state, enhancing soil nutrient circulation, and contributing to an overall improvement in the soil microenvironment [23]. Generally, biochar has a highly porous structure and high surface area, which can retain nitrogen [24,25], and thus we would assume that in both soil types, the combination of biochar and chicken manure would be beneficial with respect to reducing nitrogen leaching, increasing soil total nitrogen content, and promoting higher rice grain yield.

An important finding of this study is that the application of biochar alone would be ineffective with respect to improving rice grain yield. Despite its beneficial effects as a soil amendment, biochar is typically characterized by a relatively low nutrient composition and tends to be resistant to biodegradation, depending on production temperature, which

would limit its use as a sole nutrient source [26,27]. In this regard, our findings are consistent with those reported by Adekiya et al. [28], who found that the application of biochar would not deliver the anticipated benefits, particularly within a short time scale. Given that the high carbon content of biochar can influence the microbial decomposition of organic matter and plant nitrogen uptake, thus affecting crop yield [29]. However, Liu et al. [30] have demonstrated that the long-term application of biochar can increase total organic carbon, soil available nitrogen, phosphorus, and potassium. Furthermore, the long-term application of biochar has been demonstrated to enhance rice productivity via its effects on soil fertility and also serves as a potential measure that could contribute to mitigating greenhouse gas emissions [31]. We assume that in the future, the application of biochar alone, particularly over the long term, could contribute to increasing grain yield and improving soil chemical properties. The yield declines observed in our control treatments can be attributed to low levels of soil nutrients compared with the soils receiving amendments from the other treatments. Consequently, this would lead to an insufficient supply of N, limited carbon assimilation, and yield reductions [32].

Among the different soil amendments assessed in this study, the combined application of biochar and cow manure was observed to promote significant increases in the available P of both low- and medium-fertility soils (Figure 1), which we assume to be attributable to the high available P content (2548 mg kg$^{-1}$) in cow manure. A further noteworthy finding of the present study is that the amendment of soil with rice husk biochar and organic manures appears to contribute to the regulation of the pH and increase the EC of paddy soils, even in a soil of low fertility characterized by a particularly high pH and low EC. Compared with the control treatment, we detected significant increases in the EC of all soils receiving biochar amendment (Table 4), which we assume to be attributable to the oxidized functional groups, ash, and alkaline ions (e.g., those of Na, K, Mg, and Ca) of this material, the solubility of which can contribute to an increase in soil EC [33,34].

*4.2. CH$_4$ and N$_2$O Fluxes and Cumulative Emissions*

Rice-based cropping systems are a significant source of global anthropogenic greenhouse gas emissions, with rice production under anaerobic conditions contributing to CH$_4$ emissions via methanogenesis, whereas N$_2$O is produced as a consequence of microbial nitrification and denitrification processes [35]. In both soil types assessed in the present study, we detected the highest peak in CH$_4$ release during the latter stages of rice growth. In Japan, CH$_4$ flux during paddy rice cultivation can peak during either the early or late growing season or double peak at both stages of growth [36]. Generally, CH$_4$ fluxes from paddy rice are associated with the production, oxidation, and transportation of CH$_4$ from the soil to the atmosphere [37]. Moreover, CH$_4$ is emitted via the anaerobic decomposition of organic materials in flooded soils, with CH$_4$ escaping primarily through the diffusive transport of paddy rice plants [38].

In this study, particularly in soil amended with the combined application of biochar and cow manure, which had a high C/N ratio, the maximum CH$_4$ flux occurred during the late growing season, which we presume to be attributable to the slow decomposition of cow manure. Given that plants are unable to efficiently utilize this organic matter, a major portion of the carbon could serve as a substrate for methanogenic bacteria [39]. Our finding of an association between the C/N ratio of animal manure and methane fluxes when biochar was applied in combination with organic manures in this study is consistent with the findings of Khosa et al. [39], who demonstrated that CH$_4$ emissions were differentially enhanced depending on the C/N ratio of organic amendments. We speculate that the small CH$_4$ flux in soil treated with the combined application of 5 t ha$^{-1}$ biochar and chicken manure could be attributable to the production of fewer carbon substrates, thereby suppressing methanogenesis.

We also established that the amount of applied biochar had an effect on the cumulative CH$_4$ emissions, with higher levels being recorded from soils amended with 10 t ha$^{-1}$ biochar, applied either alone or in combination with organic manure. We speculate that

this effect could be ascribed to the carbon-rich nature of biochar and the considerable input of carbon to the soil when amended with a combination of biochar and organic manure, particularly cow manure, the most carbon-rich among the assessed treatments (Table 2). As an abundant source of carbon, biochar can also serve as a source of methanogenic substrates, thereby enhancing $CH_4$ production [40]. Moreover, the contribution of these factors would be exacerbated by the waterlogged conditions, which are particularly conducive to the decomposition of organic matter in rice paddies and thus the emission of $CH_4$. In line with expectations, the lowest $CH_4$ emissions from both soil types were detected in response to the control treatment, owing to the lack of organic input.

In contrast to $CH_4$, in both soil types, $N_2O$ emissions peaked immediately following the 2nd split of NPK application (at 35DAT), which is similar to the findings previously reported by Islam et al. [35] and Sander et al. [41]. Soil $N_2O$ emissions increase significantly in response to the addition of inorganic fertilizer, with mineral nitrogen providing sufficient substrate for nitrification and/or denitrification [42]. Our findings in the present study indicated that for both soil types, the combined application of 5 t ha$^{-1}$ rice husk biochar and chicken manure resulted in the lowest $N_2O$ emissions (Table 5). Most of the nitrogen released from chicken manure will be taken up by plants, whereas a proportion will be effectively retained by biochar [28]. Consequently, combining biochar and chicken manure is assumed to enhance the synchrony between crop nitrogen demand and soil nitrogen availability, thereby reducing $N_2O$ emissions. However, predicting the pattern of $N_2O$ emissions in response to the application of biochar and organic materials can prove difficult, as a diverse range of factors can potentially influence the release of $N_2O$ from applied organic materials, including environmental conditions (e.g., climate and soil conditions), crop conditions (e.g., crop type and residues), and management practices (e.g., type of manure, application rate, and timing) [43]. Consequently, further studies will be necessary to more precisely determine the effects of biochar on $N_2O$ production in paddy soils. Nevertheless, compared with the LF soil, we did observe a higher cumulative emission of $N_2O$ from the MF soil (Table 5). In this regard, it has been established that the type of soil is one of the factors contributing to $N_2O$ emissions, as soil characterized by a high organic matter content would typically have a higher denitrification potential, which is thus conducive to the emission of $N_2O$ [43].

**5. Conclusions**

Among the treatments assessed in this study, we established that by enhancing soil chemical properties, the combined application of 5 t ha$^{-1}$ rice husk biochar and 5 t ha$^{-1}$ chicken manure had the best synergistic effects with respect to increasing rice grain yield. Notably, regardless of soil type, the combined application of rice husk biochar and cow manure promoted a significant increase in soil available P. Furthermore, our findings revealed that methane fluxes under flooded conditions differed depending on the C/N ratio of the applied organic materials. Consequently, when combined with biochar, the C/N value of organic manures is an important property contributing to the suppression of cumulative $CH_4$ and $N_2O$ emissions. Moreover, soil type was established to be a factor contributing to greenhouse gas emissions. Although uncertainties remain regarding the efficacy of soil amendments in mitigating global warming, the balanced application of biochar combined with chicken manure was established to contribute to a satisfactory improvement in rice yield in the absence of promoting $CH_4$ and $N_2O$ emissions.

**Author Contributions:** Conceptualization, W.W.M., H.U. and Y.T.; methodology, W.W.M. and H.U; software, W.W.M.; validation, W.W.M., H.U. and Y.T.; formal analysis, W.W.M.; investigation, W.W.M. and H.U.; resources, H.U.; data curation, W.W.M. and H.U.; writing—original draft preparation, W.W.M.; writing—review and editing, H.U.; visualization, W.W.M. and H.U.; supervision, H.U.; project administration, H.U.; funding acquisition, H.U. All authors have read and agreed to the published version of the manuscript.

**Funding:** This work was supported by JSPS KAKENHI, Grant-in-Aid for Scientific Research(B), Grant Number JP22H02472.

**Institutional Review Board Statement:** Not applicable.

**Informed Consent Statement:** Not applicable.

**Data Availability Statement:** Data are contained within the article.

**Acknowledgments:** I would like to express my thanks to Amika Goto for her assistance in the gas chromatographic analysis of gas samples, as well as to all members of the Soil Sciences and Plant Nutrition laboratory at Ehime University for their assistance in laboratory work.

**Conflicts of Interest:** The authors declare no conflicts of interest.

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
