# Peer review of "Combined Effects of Rice Husk Biochar and Organic Manures on Soil Chemical Properties and Greenhouse Gas Emissions from Two Different Paddy Soils"

_soilsystems, doi:10.3390/soilsystems8010032_

Round 1

Reviewer 1 Report

Comments and Suggestions for Authors

Improving soil fertility and reducing greenhouse gas emission by organic amendments is one of very efficient measures in agricultural production. It is interesting and useful that authors have investigated the effects of rice husk biochar and organic manures on soil chemical properties and greenhouse gas emissions from two different paddy soils. In total, the MS was written sound. Hence, it is recommended to be accepted after some revisions.

1.     P1. L21, Add some more information on the results of this experiment in Abstract;

2.     P2.L84-85, give some reason why using soils of two fertility types, namely, low (LF) and medium (MF) fertility soils;

3.     P3. L102-103 give some reason why both chicken and cow manure were applied at 5 t ha-1 (10 g pot-1 on a fresh weight basis).

Comments on the Quality of English Language

Minor editing of English language was required.

Author Response

We are very grateful to the reviewer for providing a careful review, feedback, and suggestions on our manuscript. Thank you for your time and effort in reviewing our manuscript. We are pleased to receive your insightful and valuable comments. Please see the attached file. Thank you.

Reviewer 2 Report

Comments and Suggestions for Authors

Peer Review Report

1. Original submission

1.1. Recommendation

Accept in Present Form

2. Comments to Authors

Manuscript ID: soilsystems-2869909

Title: Combined effects of rice husk biochar and organic manures on soil

chemical properties and greenhouse gas emissions from two different paddy

soils

Authors: War War Mon, Yo Toma, Hideto Ueno

Overview and comments

The article represents the results of a well-planned, carefully conducted and in-depth study that deals with the changes in soil chemical properties and fertility due to a combination use of rice husks biochar and two kinds of manure. Pot experiments have been conducted to determine the optimal combinations of the biochar and chicken and cow manures concerning their effects on the cumulative N2O and CH4  emissions. The survey includes also two control samples- treatment with no biochar and with biochar alone.

In the Introduction section, based on 20 actual literary sources, a brief but meaningful analysis of the state of the art is made. It convincingly proves the need to search for alternative sustainable solutions to chemical fertilizers to improve the quality of the soil and its fertility as well as limit greenhouse emissions. The solution proposed by the authors involves the successful utilization of large-scale and renewable agricultural waste rice husks, by converting them into biochar.

The topic of the article is very important and original. Undoubtedly the results provide an advance in current knowledge dealing with natural eco-friendly soil amendments. All obtained results are represented and discussed appropriately. The conclusions are completely justified and supported by the reported results. The manuscript is written in an appropriate style and it is well organized. The layout of the experiment is designed correctly and sound technically. The used methods and protocols are described in detail. The article will be of interest to the readership of the Journal.

Author Response

(The authors gave the same response as above.)

Reviewer 3 Report

Comments and Suggestions for Authors

The article discusses the potential applications of biochar. It covers several different types of applications and provides information on industry standards and the properties of different biochars. Overall, it is well written and the information is clearly presented. Moreover, one of the challenges related to the use of biochar is the diversity of the raw material's biomass and its impact on the properties of the product. A significant impact of biochar on the growth of important nutrients in the soil has been demonstrated. Moreover, the combination of biochar, C/N value of organic fertilizers is an important property contributing to the reduction of cumulative CH4 and N2O emissions. Further research on the effect of fertilization and yielding will allow us to fully understand the mechanisms of biochar impact in the soil. Please let me know if such research/analysis is planned?

Author Response

(The authors gave the same response as above.)
